# Biological and Clinical Impacts of Glucose Metabolism in Pancreatic Ductal Adenocarcinoma

**DOI:** 10.3390/cancers15020498

**Published:** 2023-01-13

**Authors:** Zhao Liu, Hiromitsu Hayashi, Kazuki Matsumura, Norio Uemura, Yuta Shiraishi, Hiroki Sato, Hideo Baba

**Affiliations:** Department of Gastroenterological Surgery, Graduate School of Life Sciences, Kumamoto University, 1-1-1 Honjo, Kumamoto 860-8556, Japan

**Keywords:** pancreatic ductal adenocarcinoma, Warburg effect, tumor microenvironment, glucose metabolism

## Abstract

**Simple Summary:**

Pancreatic ductal adenocarcinoma is a lethal cancer with high metastatic properties. While surgical treatment is considered the first choice, the difficulty associated with early diagnosis reduces the success rates of these interventions. Therefore, early diagnosis, and thus treatment, of pancreatic ductal adenocarcinoma is extremely important. Past studies have shown that most patients with pancreatic ductal adenocarcinoma have impaired glucose tolerance, while hyperglycemia also reportedly promotes various malignant behaviors in pancreatic cancer cells. This review summarizes the clinical and molecular impacts of glycolysis on pancreatic ductal adenocarcinoma, with the aim of providing an invaluable reference for, and novel insights regarding, pancreatic ductal adenocarcinoma research and treatment.

**Abstract:**

Pancreatic ductal adenocarcinoma (PDAC) is a lethal cancer type as it is prone to metastases and is difficult to diagnose at an early stage. Despite advances in molecular detection, its clinical prognosis remains poor and it is expected to become the second leading cause of cancer-related deaths. Approximately 85% of patients develop glucose metabolism disorders, most commonly diabetes mellitus, within three years prior to their pancreatic cancer diagnosis. Diabetes, or glucose metabolism disorders related to PDAC, are typically associated with insulin resistance, and beta cell damage, among other factors. From the perspective of molecular regulatory mechanisms, glucose metabolism disorders are closely related to PDAC initiation and development and to late invasion and metastasis. In particular, abnormal glucose metabolism impacts the nutritional status and prognosis of patients with PDAC. Meanwhile, preliminary research has shown that metformin and statins are effective for the prevention or treatment of malignancies; however, no such effect has been shown in clinical trials. Hence, the causes underlying these conflicting results require further exploration. This review focuses on the clinical significance of glucose metabolism disorders in PDAC and the mechanisms behind this relationship, while also summarizing therapeutic approaches that target glycolysis.

## 1. Introduction

Pancreatic ductal adenocarcinoma (PDAC) is a malignant tumor with an extremely high mortality rate. In fact, its incidence is increasing, making it the fourth leading cause of cancer-related deaths in the USA [1]. With significant advances in clinical and biologic diagnostic techniques improvements have been noted in the overall survival rates of patients with many types of cancer. However, owing to the characteristics of PDAC, including its rapid progression, high propensity to metastasize, and resistance to various forms of anticancer therapy, significant improvements have not been made in surgical resection rates nor overall survival [2]. Approximately 80% of patients with PDAC are not candidates for surgical resection owing to progressive disease at the time of diagnosis [3]. Hence, an urgent need exists for the development of methods to improve patient prognosis.

Early PDAC leads to pancreatic duct obstruction, resulting in decreased endocrine function and induction of hyperglycemia. Previous studies have revealed that 74–88% of patients with PDAC who had diabetes were diagnosed with diabetes within 24 months prior to being diagnosed with PDAC [4,5]. Thus, long-term type 2 diabetes (T2D) may be a risk factor for the development of PDAC. Conversely, PDAC may also act as an etiological factor for T2D. However, criteria for differentiating between T2D and diabetes as an early symptom of PDAC have not been established [6]. Nevertheless, several studies have linked high blood glucose levels/T2D to the development of cancer [7,8,9,10] with most suggesting that T2D itself, not antidiabetic drugs, is a risk factor for cancer [11,12,13].

The mechanism underlying development of the association between pancreatic cancer and hyperglycemia are not well understood. It is now generally accepted that chronic inflammation, insulin resistance, hyperinsulinemia, hyperglycemia, and abnormalities in the insulin and insulin-like growth factor (IGF) axis contribute to the disease association between these two conditions [14]. In vitro studies have demonstrated that insulin can stimulate the growth of pancreatic tumor cells and that the proliferation of tumor cells is dependent on insulin dose. That is, high concentrations of insulin, combined with IGF-I, can initiate the PI3K/Akt and MAPK signaling pathways, inhibit tumor cell apoptosis, and promote cell proliferation [15,16]. Moreover, hyperglycemia increases protein kinase C expression, alters the mitogen-activated protein kinase pathway, and upregulates intercellular adhesion molecule-1 and vascular cell adhesion molecule-1 expression, which promote the epithelial–mesenchymal transition (EMT) of pancreatic tumor cells [17,18]. Under an extreme nutritional deficit and hypoxia, caused by uncontrolled growth, vascular disorders, and pro-fibrogenic responses, pancreatic cancer cells use “metabolic reprogramming” to fulfil their energy requirements and favor malignant behaviors [19,20]. Of note, pancreatic cancer cells show substantially increased glycolysis, including the overexpression of glycolytic enzymes and increased lactate production caused by mitochondrial dysfunction, onco-driver genes, specific transcription factors, a hypoxic tumor microenvironment (TME), stromal cells (e.g., cancer-associated fibroblasts [CAFs]), and tumor-associated macrophages [17,21,22,23,24].

In this review, the roles of glucose metabolism disorders in pancreatic cancer are discussed. Moreover, data from recent studies regarding the determinants and effects of a glycolytic phenotype, the effects of hyperglycemia at the molecular and clinical levels, the roles of the tumor microenvironment, and the clinical value of glycolysis as a therapeutic target, are reviewed.

## 2. Biological Role of Hyperglycemia in Cancer Cells

### 2.1. Hyperglycemia Induces Metabolic Abnormalities

Glucose can bind non-enzymatically and reactively to amino groups on proteins, lipids, and nucleic acid molecules to form precursors of advanced glycation end products (AGEs), the extent of which is proportional to the degree and duration of hyperglycemia (Figure 1) [25]. AGEs are a class of reactive molecules that destroy proteins, lipids, and nucleic acids. Their formation is part of the natural cellular senescence process, which is accelerated under high glucose conditions. Hyperglycemia leads to long-term AGE accumulation, which increases intracellular inflammatory signals—by promoting NF-κB activation—and oxidative stress, leading to cellular DNA damage and carcinogenesis [26]. Most cancer cells use aerobic glycolysis for energy uptake, rather than mitochondrial oxidative phosphorylation. Although this approach is inefficient, it facilitates the formation of nucleotides, amino acids, lipids, and other substances required for cancer cell proliferation [27]. Previous studies have shown that the aberrant expression of oncogenes/tumor suppressors (e.g., P13K, AKT, p53, and RAS) leads to the upregulation of glucose transport receptors and glycolytic enzymes, as well as translocation to the cell membrane to enhance glucose uptake [28]. Indeed, the GLUT family is upregulated in most malignancies [29]. Moreover, tumor metastasis suppressor gene-1 (MTSS1) regulates aerobic glycolysis in breast cancer by affecting HK2 kinase activity, and inhibits aerobic glycolysis in breast cancer by competitively binding to mitochondria-targeted HK2. Inhibition of aerobic glycolysis then significantly reduces the role of MTSS1 in breast cancer cell migration and invasion as well as breast cancer metastasis [30]. MicroRNA (miR)-183-5p is expressed at high levels in most cancers and is a negative prognostic predictor. Additionally, miR-183-5p may regulate aerobic glycolysis in hepatocellular carcinoma HepG2 cells via the PTEN/Akt/mTOR signaling pathway [31]. Additionally, Cai et al. demonstrated an association between Chibby expression and aerobic glycolysis in nasopharyngeal carcinoma via the Wnt/β-catenin-Lin28/let7-PDK1 cascade [32]. Meanwhile, hyperglycemia leads to activation of a specific type of protein, namely, *O*-β-*N*-acetylglucosamine (O-GlcNAc), after transferase glycosylation, which modifies intracellular proteins and acts as a signaling mechanism. The direct substrate of O-GlcNAc modification, UDP-GlcNAc, is a central component of many metabolic pathways, enabling the rapid growth and proliferation of tumor cells, which may be associated with tumor invasion and metastasis. Indeed, numerous studies have shown that the overall level of O⁃GlcNAc is increased in tumor cells [33].

### 2.2. Hyperglycemia Enhances Inflammation and Immune Dysregulation

There is evidence that hyperglycemia increases the secretion of tumor necrosis factor alpha (TNF⁃α), interferon γ (IFN⁃γ), resistin, and interleukin-6 (IL-6), leading to tumor-related inflammation [34]. These cytokines can cause mitochondrial dysfunction, oxidative stress, intracellular lipid accumulation in the liver or skeletal muscle, and reduced β⁃oxidation, resulting in insulin resistance and activation of downstream oncogenic signaling pathways, such as NF⁃κB, c⁃Jun, and JNK/MAPK [35,36,37]. In the obese population, relative hypoxia in adipocytes is caused by the inadequate perfusion of adipose tissue, increased oxygen consumption, and decoupling of adipocyte respiration. This, hypoxic condition in adipose tissues induces the upregulation of many inflammatory genes, resulting in a chronic adipose tissue inflammatory response specific to obesity [38,39]. Leukotriene B4, released from adipocytes, attracts macrophages to adipose tissue and directly attenuates insulin signaling in myocytes and hepatocytes [40]. The inhibition of inflammation-related pathways, such as NF⁃κB and JNK, as well as alteration of the expression of other signaling molecules, scaffolding proteins, and cytokines by knocking out key genes in obese mice has been shown to block hyperlipidemia-induced inflammation and insulin resistance [41]. Moreover, hyperglycemia can lead to the dysfunction of immune cells infiltrating cancer tissues, including CD8^+^ T cells, neutrophils, myeloid⁃derived suppressor cells (MDSC), and macrophages [42]. It can also affect MDSC-induced immunosuppression and tumor growth functions by reprogramming MDSC to regulate M1 and M2 differentiation [43]. In addition, hyperglycemia can stimulate monocytes and macrophages to increase IL⁃6 secretion by inducing TNF⁃α secretion, thus promoting tumor progression and infiltration (Figure 1) [44,45,46].

### 2.3. Hyperglycemia Protects Tumors against Apoptosis

Apoptosis is a form of programmed cell death that occurs in multicellular organisms and is a genetically regulated process. However, dysregulation of apoptotic mechanisms may lead to uncontrolled cell growth [47]. Hence, multiple pathways contribute toward the regulation of apoptosis. For instance, in response to multiple genotoxic stimuli, the oncogene p53 is activated and exerts its tumor suppressor function. While Serine 46 (Ser46) phosphorylation serves as a specific activator of p53, hyperglycemia specifically inhibits Ser46 phosphorylation, thereby reducing the pro-apoptotic activity of p53 [48]. During proliferation, tumor cells are in a hypoxic state with rapid metabolism. Under hypoxia, hypoxia-inducible factor 1 subunit alpha (HIF-1α) binds to other transcription factors to regulate gene transcription. Under normal aerobic conditions, HIF-1α is degraded by the prolyl hydroxylase domain (PHD). Meanwhile, hyperglycemia can regulate the stability and function of HIF-1α by interfering with its degradation by PHD enzymes, leading to enhancement of its anti-apoptotic properties in cancer cells [49]. Additionally, the pro-apoptotic activity of cytochrome C is influenced by its redox status. In cancer cells, glutathione (GSH), produced by glucose via the pentose phosphate pathway, reduces and inactivates cytochrome C. Subsequently, glucose metabolism strictly inhibits cytochrome C-mediated apoptosis (Figure 1) [50].

### 2.4. Hyperglycemia Accelerates the Malignant Behavior of Cancer Cells

In vitro studies have shown that hyperglycemia significantly enhances cell migration and invasion. Glucose can increase reactive oxygen species (ROS) production in a concentration-dependent manner and increase levels of urokinase fibrinogen activator, and superoxide dismutase-dependent hydrogen peroxide, thus, synergistically increasing cell invasion and migration [51]. Additionally, Rahn S. et al. [52] studied PDAC and found that cancer stem cells (CSC) are important contributors to tumorigenesis. That is, acquisition of CSC function is associated with the EMT, and hyperglycemia can promote the EMT and CSC properties via activation of the transforming growth factor β (TGF-β) signaling pathway, thereby contributing to tumor metastasis. Li et al. [53] reported that patients with pancreatic cancer and diabetes have higher levels of HIF-1α expression, larger tumor sizes, and deeper bile duct infiltration. Furthermore, animal studies have revealed that the endoplasmic reticulum of pancreatic follicles in diabetic mice are significantly dilated, the nuclear gap is increased, and a portion of the follicular cells contain increased chromatin near the cell membrane. Meanwhile, HIF⁃1α expression levels have been shown to increase in cancer cell lines in a glucose concentration-dependent manner. Indeed, high glucose levels might enhance tumor invasion and migration by increasing HIF⁃1α expression. Using a mouse model, Fainsod⁃Levi et al. [54] found that hyperglycemia impairs the secretion of granulocyte colony-stimulating factor (G⁃CSF) and impairs the mobilization of neutrophils, thereby increasing metastatic seeding to distant organs. Other animal experiments revealed that mice with dyslipidemia exhibit accelerated tumor growth and massive spontaneous metastasis, likely via a mechanism related to increased phosphorylation of protein kinase B Serine 473 in cancer cells by cholesterol, accompanied by reduced PI3K/Akt signaling, thus, promoting tumor cell metastasis (Figure 1) [55].

## 3. Clinical Association between PDAC and Diabetes

Approximately 80% of clinical patients with PDAC are diagnosed with weight loss that will progress to a cachectic state if left untreated [56]. Pancreatic cancer involves the disruption of intestinal nutrient absorption and glucose metabolism by influencing endocrine and exocrine function. Approximately 85% of patients with pancreatic cancer exhibit glucose metabolism disorders, with the majority (45–67%) having been diagnosed with diabetes mellitus, typically within three years before receiving their pancreatic cancer diagnosis [57,58]. However, the relationship between pancreatic cancer and diabetes is complex and the mechanism is unclear. Indeed, abnormal glucose metabolism is closely related to the nutritional status and prognosis of patients with pancreatic cancer.

In a 10-year follow-up of more than 90,000 people, Inoue et al. found that the incidences of pancreatic, liver, and kidney cancers were significantly higher in patients with diabetes than in those without diabetes [59]. In another study of more than 30,000 individuals, the incidence of pancreatic cancer in diabetic patients was 2.5 times higher than that in non-diabetic patients [60]. In this study, the incidence of esophageal and colorectal cancers was significantly higher in diabetic patients than in controls. The blood glucose range affects the incidence of different cancers to some extent, suggesting that the development of malignancies due to diabetes is caused by chronic hyperglycemia or hyperinsulinemia. Stolzenberg et al. showed that hyperglycemia, hyperinsulinemia, and insulin resistance are all risk factors for pancreatic cancer; however, the effects of these factors on the development of pancreatic cancer were not statistically significant [61].

In a study by Mizuno et al., the median survival time for pancreatic cancer was improved by approximately 10 months (20.2 months vs. 10.2 months) in patients with new onset diabetes mellitus alone compared to patients without diabetes mellitus [62]. This improved prognosis may be associated with the early diagnosis and treatment of pancreatic cancer due to diabetes mellitus. Therefore, the identification of new-onset diabetes is of immense clinical importance for the early diagnosis and management of pancreatic cancer. However, long-term diabetes has been found to be a poor prognostic factor for patients with pancreatic cancer, while postoperative glycemic recovery in patients with new-onset diabetes is a protective prognostic factor. Hence, glucose metabolism disorders are highly related to pancreatic cancer prognosis [63].

Diabetes mellitus may contribute to pancreatic cancer prognosis through various potential mechanisms. First, T2D is often associated with insulin resistance, unstable insulin concentrations, reduced ability of the liver to inhibit inappropriate hepatic glycogen release, and reduced ability of β-cells to overcome insulin resistance. However, the causes of β cell dysfunction are complex. Typically, β cell dysfunction is caused by an elevated glucose concentration due to insufficient glucose sensing to stimulate insulin secretion. Glucose concentrations that remain persistently higher than the normal physiological range can lead to the manifestation of hyperglycemia [64].

Moreover, diabetes may act as a contributor to pancreatic cancer development, preceding the diagnosis of cancer. This may be related to several mechanisms. First, the release of exosomes that deliver adrenomedullin to β-cells induces endoplasmic reticulum stress and perturbation of the unfolded protein response, leading to β-cell dysfunction and death [65]. Second, inflammatory responses mediate β-cell dysfunction; the abnormal activation of oxidative stress and the unfolded protein response leads to a reduction in β-cells and subsequent apoptosis and senescence [35,66]. Third, T2D is often concurrent with obesity, which represents another high-risk factor for pancreatic cancer [67]. These potential mechanisms can lead to increased insulin levels in the pancreatic microenvironment and further contribute to tumor development.

Pancreatic-derived diabetes mellitus, also known as type 3c diabetes mellitus (T3cD), typically occurs secondary to hereditary or acquired pancreatic disease or following pancreatectomy. T3cD differs from traditional T1D or T2D in that it is often caused by exocrine pancreatic injury, such as infection, acute or chronic pancreatitis, trauma, or surgery resulting in reduction of islet cells; its main symptoms include hypoinsulinemia, hepatic insulin resistance, decreased pancreatic polypeptide levels, and increased peripheral insulin sensitivity. T3cD is also associated with pancreatic cancer. Pancreatic cancer-associated diabetes has unique hormonal and metabolic features. That is, insulin levels may be mildly elevated and peripheral insulin resistance occurs, which more closely resembles T2D than T3cD of other etiologies [68]. In cases of chronic pancreatitis, pancreatic cancer, and pancreatic resection, pancreatic polypeptide (PP) is also deficient. Indeed, the amount of PP—primarily produced in the head of the pancreas—plays an important role in pancreatic-derived diabetes. Therefore, the ADA guidelines specifically indicate that pancreatic cancer-associated diabetes differs from other T3cD in that it may occur by mechanisms other than a decrease in the number of islet β cells [69]. The above research advances may improve the diagnostic strategy of pancreatic cancer.

## 4. Promotion of PDAC through a High Glycolytic Phenotype

While differentiated cells rely primarily on mitochondrial oxidative phosphorylation to generate energy for physiological activities, pancreatic cancer cells rely on aerobic glycolysis, a phenomenon known as the “Warburg effect” [70,71,72]. Aerobic glycolysis is an inefficient way to produce ATP; however, it facilitates the formation of substances required for pancreatic cancer cell proliferation, such as nucleotides, amino acids, and lipids [73]. In this regard, hyperglycemia can provide an adequate source of glucose for rapidly proliferating pancreatic cancer cells. The aberrant expression of oncogenes/tumor suppressors (e.g., PI3K, AKT, p53, and RAS) results in increased levels of glucose transporters (GLUTs) and glycolytic enzymes at the cell membrane, as well as increased glucose uptake [28,74]. In particular, Chikamoto et al. reported that GLUT1 is an unfavorable prognostic factor in pancreatic cancer [75].

In PDAC, distant metastasis is typically associated with a poor prognosis [76]. The glycolytic phenotype can promote the EMT of cancer cells, thus, enhancing aggressiveness and promoting metastasis [7,77,78]. Induction of the glycolytic phenotype of PDAC is regulated by multiple pathways [79]. More specifically, increases in the expression of oncogenes [80], mitochondrial dysfunction [81,82], and the abnormal expression or activation of glycolytic enzymes all contribute to the enhanced glycolytic phenotype in PDAC [83,84].

Warburg noted that a deficiency in irreversible oxidative phosphorylation (OXPHOS) can lead to aerobic glycolysis. Moreover, a decrease in the OXPOHS energy supply is often accompanied by structural abnormalities in mitochondrial DNA (mtDNA) [22,85]. Alterations in mitochondrial function caused by mtDNA mutations or copy number changes, as well as defective mitochondrial respiratory chain complexes and ROS, trigger retrograde signals from the mitochondria to the nucleus. When the mtDNA copy number is reduced, membrane potential (ΔΨm) is disrupted, which mimics Ca^2+^-regulated neurophosphatase signaling and activates the upregulation of many oncogenic factors and kinases, such as IGF-1R/NF-κB/PI3K, and glycolytic enzymes [83,84,86] (Figure 2). In contrast, glucose restriction inhibits various energy-dependent pathways in pancreatic cancer cells, such as IGF-1/PI3K/Akt/mTOR, inhibits cell metabolism and growth, and promotes G1 phase block and apoptosis [87,88].

Furthermore, fluctuations in the levels of ROS produced by mitochondria induce mtDNA mutations and enhance the metastatic potential of cancer cells [89,90]. In fact, reduction of NAPDH oxidase expression—a specific enzyme that regulates ROS production—inhibits the glucose uptake capacity of PANC-1 cells, and attenuates tumor growth in vivo [91,92,93].

## 5. The TME in Pancreatic Cancer

The TME refers to the conditions in which tumor cells improve growth via the secretion of various factors and the promotion of paracrine secretion. Conversely, systemic and local tissues can be altered by immune, secretory, and metabolic modalities to limit the influence of these factors on tumor development. The homeostasis and evolution of the TME is controlled by close crosstalk within and between cells, including malignant cells, endothelial cells, stromal cells, and immune cells. This complex interaction not only constitutes a source of energy supply but also acts as a mechanism of communication between different cellular compartments [94].

Cancer cells use metabolic byproducts to hijack the function of tumor-infiltrating immune cells for their own benefit. This applies to the glycolytic cancer cell secretion of lactate, which typically favors immune cell polarization to an immunosuppressive phenotype [95,96,97]. As a highly malignant tumor, pancreatic cancer is hypoimmunogenic and highly immunosuppressive, causing associated immunotherapies to have minimal clinical success, likely limited in large part by the immunodeficiency and immunosuppression of the TME. Immune cells in the pancreatic cancer TME are divided into three main categories: tumor-associated macrophages, tumor-infiltrating T lymphocytes, and myeloid-derived suppressor cells. In pancreatic cancer, tumor-associated macrophages are a marker of poor prognosis and tumor cells interact with macrophages via IL-13 to induce their differentiation into the M2 phenotype, which not only promotes tumor growth, metastasis and angiogenesis [98,99], but also recruits regulatory T lymphocytes (Tregs) and inhibits the cytotoxic effects of effector T lymphocytes on tumors [100]. T lymphocytes are prevalent in the pancreatic cancer stroma, however, the activity of effector T lymphocytes, which have a major anticancer role, is greatly limited. In fact, Th2, M2 macrophages, Tregs, and myeloid-derived suppressor cells, which are abundant in pancreatic cancer, can block the antitumor response of Th1 and CD8^+^ T lymphocytes to promote tumor growth. Meanwhile, patients with pancreatic cancer that have high Th1 and CD8^+^ T lymphocyte counts within the TME typically have a better prognosis [98,100]. Moreover, PD-1 is present on the surface of T lymphocytes, and T lymphocyte receptors recognize the major histocompatibility complex on the surface of cancer cells. T lymphocytes are activated to secrete IFNγ, subsequently, high local concentrations of IFNγ and TNFα secreted by tumor-associated macrophages induce cancer cells to express programmed death ligand 1, which specifically binds PD-1 and initiates T lymphocyte apoptosis, thereby enabling cancer cells to achieve immune escape [101,102]. Moreover, myeloid-derived suppressor cells are tumor cells that induce the differentiation of immature myeloid cells, inhibit the proliferation and differentiation of CD4^+^ T lymphocytes and CD8^+^ T lymphocytes, limit the immune surveillance of CD8^+^ T lymphocytes, and promote the expansion of Tregs to assist the immune escape of tumor cells [103].

One of the most striking features of PDAC is the abundance of stromal cells [104]. Enhanced extracellular matrix stiffness in PDAC can promote chemoresistance as well as malignant tumor behavior. Moreover, Li et al. reported that enhanced ECM stiffness can promote the glycolytic capacity of CAFs by upregulating YAP/TAZ expression [105]. Indeed, the inhibition of protein interacting with never in mitosis A1 (Pin1), the overexpression of which in PDAC is correlated with the desmoplastic and immunosuppressive TME and poor patient survival, simultaneously blocks multiple cancer pathways, disrupts pro-connective tissue proliferation and the immunosuppressive TME, and upregulates programmed cell death protein ligand (PD-L1) and equilibrative nucleoside transporters (ENT1), allowing PDAC to be eradicated by immunochemotherapy [106].

Notably, aerobic glycolysis in CAFs results in the production of high energy metabolites (e.g., lactate and pyruvate). These are transferred to adjacent epithelial cancer cells and then passed through aerobic mitochondrial metabolic processes within cancer cells to produce ATP and promote tumor growth and metastasis, a process known as the “reverse Warburg effect” [107]. This glycolytic phenotype in stromal cells (e.g., CAF) supports the survival of neighboring cancer cells and increases OXPHOS activity and invasiveness [108]. Thus, stromal cells—that constitute the majority of pancreatic tumors—can promote cancer cell growth and progression by promoting the glycolytic phenotype and supporting mitochondrial metabolism in oxidative PDAC cells. Hence, targeting the stroma of PDAC is a promising anticancer strategy.

## 6. Therapeutic Strategies Targeting Glycolysis

A basic approach to preventing the development of malignant tumors in patients with abnormal glucose metabolism is to prevent risk factors, i.e., to control blood glucose concentrations and to actively treat obesity, diabetes, and hypertension. Meanwhile, the risk of malignancy in the healthy population can be reduced by correcting, or improving, abnormalities in glucose metabolism by lifestyle changes, including weight loss and adoption of a healthy diet (Table 1) [109]. Similar results can be achieved through surgery; for example, obese patients that have undergone bariatric surgery have a lower incidence of malignant tumors than that of the control group, comprising obese patients without surgery (Table 1) [110].

Metformin has a strong protective effect against myriad malignancies in the diabetic population [14]. Although most basic experimental studies have shown some degree of protection against certain malignancies (e.g., breast, colorectal, and prostate cancers), recent clinical trials and meta-analyses have yielded inconsistent findings (Table 1) [111]. Hence, further studies are needed to determine whether metformin can be developed as a complementary therapy to other anticancer therapies. Moreover, cardamonin has been shown to inhibit the growth of a triple-negative breast cancer cell line, MDA-MB-231, in vitro and in vivo by inhibiting HIF-1α-mediated cellular metabolism. In addition, cardamonin treatment reduces glucose uptake and lactate production and efflux, suggesting that it contributes to the inhibition of the glycolytic process (Table 1) [112]. Meanwhile, Yan et al. demonstrated the critical role of glucose metabolism in KRAS-driven PDAC resistant to MAPK inhibition, revealing potential therapeutic approaches to treat this aggressive disease (Table 1) [113].

It is important to note that glucose metabolism disorders caused by pancreatic cancer commonly result in exocrine pancreatic dysfunction. A study of 248 patients has shown that up to 92% of patients with pancreatic exocrine abnormalities, with various causes (pancreatic cancer or chronic pancreatitis), exhibit vitamin D deficiency (Table 1) [114]. The level of vitamin D in the body is correlated with blood glucose control. Hence, monitoring and supplementation of 25-hydroxyvitamin D levels is beneficial for treatment. The function and physiological levels of enteroglycemic hormone in normal beta cells depends on the exocrine function of the pancreas and sound lipolysis. Hence, impaired function of the enteroglycemic system plays a role in glucose metabolism dysfunction in pancreatic cancer, and the maintenance of the enteroglycemic system function may be a therapeutic target [115].

In conclusion, the dysfunction of glucose metabolism is closely related to the development of malignant tumors, and the associations between various components of abnormal glucose metabolism and different types of malignant tumors vary. Abnormal glucose metabolism is both a risk factor for malignant tumor development and an accelerant of tumor development that inhibits tumor cell apoptosis and enhances tumor cell metabolism, proliferation, and metastasis, among other processes. Although a variety of molecular mechanisms have been investigated, conflicting experimental findings have been reported on related therapeutic approaches. For instance, basic research has shown that metformin and statins are effective for the prevention or treatment of malignancies; however, no such effect has been found in clinical trials, and the specific reasons for these conflicting results warrant further exploration.

## 7. Conclusions and Future Directions

Patients with pancreatic cancer are prone to glucose metabolism disorders, and diabetes is closely related to the development of pancreatic cancer. The mechanisms underlying these relationships are complex. Patients with diabetes have a significantly higher risk of developing pancreatic cancer than do healthy individuals, and patients with pancreatic cancer often have glucose metabolism disorders or diabetes mellitus. Diabetes or glucose metabolism disorders associated with pancreatic cancer are typically associated with insulin resistance, beta cell damage, and other factors. Considering that one of the early manifestations of pancreatic cancer may be new onset diabetes mellitus, the differential diagnosis of new onset diabetes mellitus must be seriously considered. From the perspective of molecular regulatory mechanisms, glucose metabolism disorders are closely related to the initiation and development of pancreatic cancer to late invasion and metastasis. It is, therefore, important to closely monitor glucose metabolism disorders in pancreatic cancer for early detection to improve efficacy of treatment strategies and overall prognosis.

## Figures and Tables

**Figure 1 cancers-15-00498-f001:**
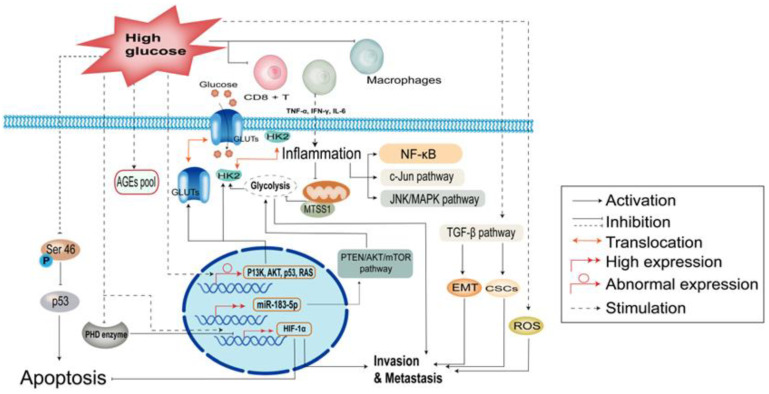
Mechanisms by which abnormal glucose metabolism contributes to malignant tumor development.

**Figure 2 cancers-15-00498-f002:**
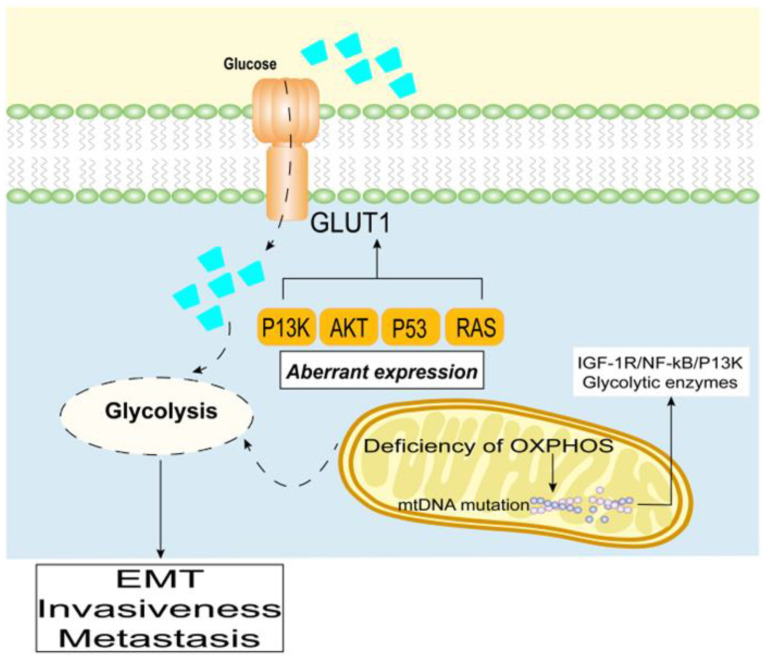
Promotion of PDAC through a high glycolytic phenotype.

**Table 1 cancers-15-00498-t001:** Therapeutic strategies targeting glycolysis.

Therapeutic Agent	Therapeutic Strategy	Cancer type	Reference
Prevent risk factors	Weight loss and the adoption of a healthy diet, surgery	Most of malignant tumors	[109,110]
Metformin, sulfonylureas, thiazolidinediones (pioglitazone)	Control the blood glucose	Breast, Pancreatic, Colorectal, and Prostate cancers	[14,111]
Cardamonin	Inhibit HIF-1α, inhibition of the glycolytic process	Breast cancer	[112]
Trametinib, GDC-0623	Inhibit the MEK pathway	PDAC	[113]
Vitamin D	Monitoring and supplementation of 25-hydroxyvitamin D levels is beneficial for treatment	Pancreatic cancer	[114]

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
