# Peer review of "Biological and Clinical Impacts of Glucose Metabolism in Pancreatic Ductal Adenocarcinoma"

_cancers, 2023, doi:10.3390/cancers15020498_

Round 1

Reviewer 1 Report

In this review, the authors comprehensively talked the glucose metabolism and the PDAC cancer. But it can not be accepted before revisions. Here are the comments and suggestions:

1.      In the abstract, the authors informed much information on the background while few information on the content and conclusions, it is better to adjust it.

2.      In part2, Promotion of PDAC through a high glycolytic phenotype, the authors informed that the PDAC reply on more on glycolysis than OXPHOS and a series of pathways in both energy sources are discussed here. Please add a figure for easier understanding.

3.      In part 3, the author summarized three mechanisms by which diabetes mellitus contributing to the PDAC, the author mentioned β-cell dysfunction in the second point, please inform what is the β-cell function and what are the relations between this cell’s function and PDAC.

4.      Figure 2: Can you draw it within a cell, so that we can see which pathway work in mitochondria and which in cytostome. This figure is too simple which can not reflect the information mentioned in part 4.

5.      In part 2, the author discussed glycolysis and PDAC and in Part 4, the author discussed glycolysis and cancer, it seems you need to introduce the function of glycolysis in cancer first since it is broader, and then narrow down to PDAC.

6.      Part 4 is listed in a very big volume, I suggest you add sub-titles under this part since you are mentioning several different biological processes. Thus, it will be more readable and attractive, easy for readers to grasp the key idea.

7.      Part 5 is the TME in PDAC, can you link it to the title of this review? Since the title is focusing on the glucose metabolism, so it is better to discuss the interplay of glucose metabolism and immunity.

8.      In part 6, therapeutic strategies targeting glycolysis, is it possible to make a table listing all the antibodies and small molecules targeting glycolysis? If yes, please add it. Please also emphasis the ones which are already on clinical trials.

9.      The references the author cited are too old, please add the latest references and content in the text, especially within the recent three years.

Author Response

  1. In the abstract, the authors informed much information on the background while few information on the content and conclusions, it is better to adjust it.
  • Thank you so much for your kindly recommendation, we have modified the abstract according to your suggestion, please check it.

  1. In part2, Promotion of PDAC through a high glycolytic phenotype, the authors informed that the PDAC reply on more on glycolysis than OXPHOS and a series of pathways in both energy sources are discussed here. Please add a figure for easier understanding.
  • Thanks a lot for your suggestion, we made a brief figure (Figure 2), it would be great if you could check it out.

  1. In part 3, the author summarized three mechanisms by which diabetes mellitus contributing to the PDAC, the author mentioned β-cell dysfunction in the second point, please inform what is the β-cell function and what are the relations between this cell’s function and PDAC.
  • Thank you for your suggestion, we have added the content about β-cell to that section, please check it out.

  1. Figure 2: Can you draw it within a cell, so that we can see which pathway work in mitochondria and which in cytostome. This figure is too simple which can not reflect the information mentioned in part 4.
  • Thanks for your suggestion, we modified that figure (Figure 1), please check it.

  1. In part 2, the author discussed glycolysis and PDAC and in Part 4, the author discussed glycolysis and cancer, it seems you need to introduce the function of glycolysis in cancer first since it is broader, and then narrow down to PDAC.
  • Thank you very much for your suggestion, we have adjusted the order of these two parts, please check it.

  1. Part 4 is listed in a very big volume, I suggest you add sub-titles under this part since you are mentioning several different biological processes. Thus, it will be more readable and attractive, easy for readers to grasp the key idea.
  • Thank you very much for your suggestion and we have made the changes as you proposed.

  1. Part 5 is the TME in PDAC, can you link it to the title of this review? Since the title is focusing on the glucose metabolism, so it is better to discuss the interplay of glucose metabolism and immunity.
  • Thank you very much for your valuable comments. Regarding this section, we would like to express that the TME, as a research hotspot, has been found to be linked to glycolysis in many current basic studies (including our own project). Therefore, we have included the tumor microenvironment in this section.

  1. In part 6, therapeutic strategies targeting glycolysis, is it possible to make a table listing all the antibodies and small molecules targeting glycolysis? If yes, please add it. Please also emphasis the ones which are already on clinical trials.
  • Thank you so much for your recommendation, we added a table about this section.

  1. The references the author cited are too old, please add the latest references and content in the text, especially within the recent three years.
  • Thank you so much for your suggestion, we updated some latest references.

Reviewer 2 Report

The present review aimed at discussing the role of biological and clinical role of glucose in pancreatic cancer; a deadliest tumour with extensive metabolic reprogramming

My comments:

          The authors aimed at dissecting the role of glucose metabolism in PDAC, but keep using pancreatic cancer and PDAC interchangeably, the authors need to confine the review specifically to PDAC or more generally to pancreatic cancer; in both cases the authors need to revised the review appropriately

          The section 4: the biological role of glycolysis in cancer, Why did you discuss this in general to all cancers while you discussed the section before just in pancreatic cancer. This is a major limitation which needs to be addressed; the authors need to revise the review more comprehensively in relation to PDAC.

          The figures are not of much scientific values and quality, and should be improved.

          The review need to undergo significant smoothing for the English to make it easy to read and understand.

          The following key papers should be cited:

https://www.mdpi.com/2072-6694/11/10/1460

https://www.nature.com/articles/s41419-022-05259-w

https://doi.org/10.1158/0008-5472.CAN-20-3792

Author Response

  • The section 4: the biological role of glycolysis in cancer, Why did you discuss this in general to all cancers while you discussed the section before just in pancreatic cancer. This is a major limitation which needs to be addressed; the authors need to revise the review more comprehensively in relation to PDAC.
  • Thank you very much for your questions and suggestions. We apologize that in the previous manuscript we put the pancreatic cancer discussion before the cancer discussion and we have made changes. The adjusted logic then reads: introduce the function of glycolysis in cancer first since it is broader, and then narrow down to PDAC.
  • The figures are not of much scientific values and quality, and should be improved.
  • Thank you so much for your suggestion. We modified the Figure and added another new figure in the manuscript.
  • The review needs to undergo significant smoothing for the English to make it easy to read and understand.
  • Thank you very much for your suggestions. com has done a second round of English calibration for our manuscript.
  • The following key papers should be cited:

https://www.mdpi.com/2072-6694/11/10/1460

https://www.nature.com/articles/s41419-022-05259-w

https://doi.org/10.1158/0008-5472.CAN-20-3792

  • Thank you very much for your suggestion and we have cited these references in the manuscript for your confirmation. The first article cited as No. 10, the second was cited as No. 74, the third cited as No. 113.

Round 2

Reviewer 1 Report

All the required revisions were did in the revision manuscript. I suggest to accept.

Reviewer 2 Report

The authors has appropriately addressed the issues raised in my first report. Hence the present review may be accepted for publication in cancers after proper editing and formating